# Harmonizing Nature, Education, Engineering and Creativity: An Interdisciplinary Educational Exploration of Engineered Living Materials, Artistry and Sustainability Using Collaborative Mycelium Brick Construction

**DOI:** 10.3390/biomimetics9090525

**Published:** 2024-08-31

**Authors:** Richard W. van Nieuwenhoven, Matthias Gabl, Ruth Mateus-Berr, Ille C. Gebeshuber

**Affiliations:** 1Institute of Applied Physics, TU Wien, 1040 Vienna, Austria; gebeshuber@iap.tuwien.ac.at; 2Center for Didactics of Art and Interdisciplinary Education, University of Applied Arts Vienna, 1010 Vienna, Austria; matthiasgabl89@gmail.com (M.G.); ruth.mateus-berr@uni-ak.ac.at (R.M.-B.)

**Keywords:** engineered living materials, artistry, sustainability, mycelium construction materials, interdisciplinary education

## Abstract

This study presents an innovative approach to interdisciplinary education by integrating biology, engineering and art principles to foster holistic learning experiences for middle-schoolers aged 11–12. The focus lies on assembling mycelium bricks as engineered living materials, with promising applications in sustainable construction. Through a collaborative group task, children engage in the hands-on creation of these bricks, gaining insights into mycology, biomaterials engineering and artistic expression. The curriculum introduces fundamental concepts of mycelial growth and its potential in sustainable material development. Children actively participate in fabricating 3D forms (negative and positive) using mycelium bricks, thereby gaining practical knowledge in shaping and moulding living materials. This hands-on experience enhances their understanding of biological processes and cultivates an appreciation for sustainable design principles. The group task encourages teamwork, problem-solving and creativity as children collaboratively compose structures using mycelium bricks. Integrating art into the activity adds a creative dimension, allowing participants to explore aesthetic aspects while reinforcing the project’s interdisciplinary nature. Conversations about the material’s end-of-life and decomposition are framed within the broader context of Nature’s cycles, facilitating an understanding of sustainability. This interdisciplinary pedagogical approach provides a model for educators seeking to integrate diverse fields of knowledge into a cohesive and engaging learning experience. The study contributes to the emerging field of nature-inspired education, illustrating the potential of integrating living materials and 3D-understanding activities to nurture a holistic understanding of science, engineering and artistic expression in young learners.

## 1. Introduction

Ruth Mateus-Berr and Ille C. Gebeshuber initiated an interdisciplinary workshop hosted at the University of Applied Arts Vienna. This workshop served as a platform for participants to engage in discussions centred around integrating art and science within a middle-school curriculum with a specific emphasis on inspiration and sustainability from Nature. Richard W. van Nieuwenhoven and Matthias Gabl collaborated on a course syllabus to guide middle-schoolers aged 11 to 12 in creating artwork utilizing engineered living materials (ELMs). Given their well-established research foundation and their renowned living component, mycelium-based ELMs emerged as the logical choice for this endeavour [1].

In recent years, there has been a remarkable surge in the exploration and application of mycelium materials, particularly in packaging and construction [2]. This growing interest signifies a notable shift towards harnessing the potential of ELMs as viable alternatives in various industries [1,3]. While the prevalent practice involves subjecting the mycelium to heat treatment during the post-growth phase to deactivate its biological activity, exploring methods that leave it alive could open new possibilities for dynamic and self-healing materials. The accessibility of mycelium production methodologies has democratised its usage. Indeed, the emergence of do-it-yourself (DIY) kits [4], made available by various companies, has facilitated widespread experimentation and innovation in this visionary field. Central to the appeal of mycelium-based ELMs is their inherent sustainability [5], which resonates deeply with the ethos of contemporary environmental commitment [6,7]. Mycelium offers a renewable and biodegradable alternative, unlike traditional materials derived from finite resources. Utilising local resources for the materials and producing them on-site further bolsters their eco-credentials, contributing to the promotion of circular economy principles [8]. Moreover, mycelium-based materials exhibit remarkable versatility in their application. A combination of moulding, casting and shaping techniques can be tailored to meet a diverse range of functional and aesthetic requirements. This adaptability opens up various possibilities across industries, from customisable packaging solutions to structurally sound architectural elements [9,10]. Beyond their practical utility, mycelium materials are a potent educational tool, offering a multifaceted learning experience that transcends traditional disciplinary boundaries. The convergence of scientific principles, sustainable practices and artistic expression presents a unique opportunity for interdisciplinary engagement, especially among young learners, by integrating biology, material science, sustainability and craftsmanship concepts into educational initiatives centred around mycelium materials, fostering a holistic understanding of complex scientific concepts while nurturing creativity and critical thinking skills [11,12,13].

Interdisciplinarity “has been linked with attempts to expose the dangers of fragmentation”. In a century where knowledge in science doubles every two years, interdisciplinarity should uncover and develop connections between disciplines, “re-establish old connections” and create innovative solutions [14]. Klein argues that the basic structure of interdisciplinarity has been, since its beginning, an “inseparable implication of disciplinary thought” ([15], p. 20). In her opinion ([14], p. 188), an interdisciplinary approach is a process for achieving an integrative synthesis, “a process that usually begins with a problem, question, topic or issue”. The essential part is problem-solving through transgressing disciplinary language and worldviews. Klein [14], pp. 188–189, described steps: defining, determining, developing, specifying, engaging, gathering, resolving, building and maintaining, collating, integrating, confirming or disconfirming and deciding. These steps are similar to the steps of the Design Thinking methodology, which are supplemented by teamwork [16]. This course syllabus extends the valuable interdisciplinarity insights provided by Yeter et al. [17] and Coban and Coştu [18] by emphasizing the hands-on use of mycelium-based living materials, highlighting sustainability and fostering a practical appreciation for renewable and biodegradable alternatives in engineering and design.

Design Thinking is seen as an essential way to cultivate 21st-century skills, according to (Li and Zhan [19], p. 8), who conducted a meta-study of the application of Design Thinking (43 SSCI peer-reviewed journal papers with 44 studies) in the school context. The authors note that the need and interest in introducing primary to secondary schools to Design Thinking has increased by almost 30% since 2017. Primarily, the method is used for science, technology, engineering and mathematics courses (STEM) in the younger grades [20]. Interestingly, this method was hardly used in subjects such as technology and design, but rather, it was used in science and engineering. Art and design education must re-claim the methodology.

Considering the compelling attributes elucidated, this article provides educators with a structured approach for incorporating an interdisciplinary approach using mycelium-based living materials into middle-school science and art education. Klein [14] argues that interdisciplinarity combats fragmentation by developing connections between disciplines, promoting problem-solving and creating innovative solutions. This process involves steps similar to Design Thinking, such as defining, developing, engaging and deciding, enhanced by teamwork [16].

## 2. Materials and Methods

During the specified interdisciplinary workshop hosted at the University of Applied Arts Vienna, Richard W. van Nieuwenhoven and Matthias Gabl initiated the design of an interdisciplinary middle-school course syllabus centred around a sculpture made from mycelium bricks.

### 2.1. Workshop Setup

To encompass a wide range of disciplines, the curriculum needed to incorporate mathematics, perspective drawing, craftsmanship, sustainability, mould forming and laboratory handling of living materials, all within an engaging and playful learning environment. As the provided study also should foster 3D visualisation [21], the decision was made to make all participants’ workpieces fit each other. For this, a solution was found to use mycelium basic blocks of 4 × 4 × 4 cm^3^ as a basis.

The course syllabus begins with students crafting a mould for a designed clay form, which is then used to cultivate mycelium. The resulting mycelium forms are subsequently integrated into a collective sculpture. The instructors include the school’s craft and art teacher, Matthias Gabl, and a scientist, Richard W. van Nieuwenhoven, who elucidate the scientific concepts to the schoolers. The 13 schoolers (6 boys and 7 girls) are in middle-school (using the second-grade gymnasium in the Austrian system, ages 11–12) and begin the course without prior preparation, except for from the biology teacher, who previously covered the topic of mushrooms. It was decided that schoolers would be informed incrementally, as explaining the entire procedure in a single session was anticipated to be too challenging.

The course is held at a gymnasium (Austrian education system [22]) located in a village near Vienna in blocks of 90 min with one-week intervals. The exact location and identities of the students is not disclosed to protect the schoolers personal rights.

### 2.2. Design of the Flat Form

Participants must design a flat form (FF) constructed from four basic blocks. Mathematically, there are just three possible forms; this is utilised in the first class session to illustrate that many of the individual forms are, in reality, equal to each other after spacial rearrangement (Figure 1 and Figure 2). This first class session also teaches the participants to think in the third dimension and to express these thoughts on paper as drawings [21].

A personalized marking is included in the design to identify each individual FF in the shared structure built at the course’s end.

### 2.3. Construction of the Flat Form

The second task for the participants is to realise the designed form in clay [23]. In this class, the teacher instructs the participants on the clay-forming methods so that the proportions are according to the design. Personalized markings are added using a scraper on top of the structure. Because the different FFs are interconnected later in the course, the teacher has to verify the proportions of every participant.

During this phase, the teacher has already told the middle-schoolers that the structures shall be combined later, so the participants can think about the different possibilities for connecting the individual pieces.

### 2.4. Construction of the Negative Flat Form

The third task is to create a negative of the constructed clay form (mould). This is done by utilising casting gypsum. By forming a clay wall around the FF of around 5 cm height, space is created for casting gypsum. To reduce the gypsum needed for casting, the space between the clay wall and the clay structure should be around 1 cm wide (Figure 3).

The participants need to collect the right paster ingredients and mix them appropriately. The cast-ready gypsum is then poured into the prepared clay form. Depending on the product used, this cast must dry according to the gypsum specifications (Figure 3). In this study, we utilise PUFAS Modellgips (PUFAS, Leipzig, Germany) and need a resting period of at least one day.

### 2.5. Mycelium Tetris Block

The fourth task starts with carefully separating the FF from the negative gypsum form (Figure 4). After cleaning the FF, the DIY mycelium is needed; it is best to contact the company that sells the DIY kit in person to synchronise the arrival of the DIY with the start of this phase in the class. Because the mycelium is very sensitive to bacterial infections, it must be handled as sterilely as possible. Consequently, lab gloves and masks were a requirement for all participants. The sterile handling of the materials also adds extra suspense to the procedure, which helps to keep the participants interested. The gloves and the FF must be sterilised, which is done with 70% isopropanol. The teacher hands out the isopropanol and the disinfection process is done in front of the teacher to keep the participants from misusing it. Instructions are given not to touch anything that is not sterilised.

The sterile gear provided is a teaspoon, a small container for the flour, a container for the mycelium-impregnated wood spans, two cling film pieces of at least 20 × 20 cm^2^ and two toothpicks. The participants collect 2 g of flour and around 70 g of mycelium-impregnated wood spans with these utensils.

Back at their workplaces, the participants first carefully (by not touching the FF with their gloves, as the FF is likely not fully sterile) fill the gypsum form with the cling film. The flour and the mycelium-impregnated wood spans are then well-mixed in the mycelium container. So much of the mixture is now cast into the FF that it is filled, and every time the layer is one centimetre higher, the layer is pressed with glove-covered fingers into the edges of the form. The teacher prepares a sample for the best results so that the participants can see the procedure before practising it. As the FF is filled to the top, it is covered with the second cling film, which is perforated every 3 cm using toothpicks.

### 2.6. Incubation

The filled FFs of all participants are then placed together to incubate for five days, whereas seven should also be possible, but that would probably terminate the growth phase and would not allow for the merged growth of the shared structure. The ideal temperature for incubation is 24 °C, depending on the DIY kit utilised.

The incubation place is best selected for minimal interaction, as the mycelium is very sensitive during the initial growth. Nevertheless, for regular checking by the participants, the incubation place should be visible: for example, in a small glass house (Figure 5).

### 2.7. Shared Mycelium Construction

After five days, the mycelium should have grown to cover the wood spans with a whitish-like texture (Figure 6). The cling film allows easy extraction from the FF. During extraction, the participants must decide on the three-dimensional shared form built from the individual bricks. Again, gloves with disinfection procedures are used. Disinfected toothpicks are utilised to connect the mycelium Tetris blocks. One week of additional incubation at the same incubation place allows the material to grow together. Note that this only happens sometimes and strongly depends on many individual parameters.

### 2.8. Decomposition Process

The last step of this course is the exposure of the structure to a natural environment to view the decomposition of the material and its integration back into Nature. It is recommended to visit the place of last year’s course with the participants of the following year to view the progress of the decomposition.

### 2.9. Educational Objectives and Competencies

The objectives and competencies align with the curriculum guidelines for “Technology and Design” and “Art and Design” set by the Austrian Federal Ministry of Education for general secondary schools (*Lehrpläne—allgemeinbildende höhere Schulen des Bildungsministerium für Bildung für die Unterrichtsfächer, Bundesministerium für Bildung 2023* [22]).

Children who learn to collaboratively three-dimensionally model acquire diverse competencies and skills encompassing cognitive, motor and socio–emotional domains [24,25,26].

#### 2.9.1. Fine Motor Skills

Learning to model in three dimensions fosters children’s fine motor skills development. This skill development is evident as they manipulate clay, modelling compounds and basic blocks [27]. Over time, their precision in movements and attention to detail improves significantly.

#### 2.9.2. Creativity and Experimentation

Children who engage in different imaginary modelling techniques develop a sense of creativity and an eagerness to experiment [28]. They explore how different forces, shapes and movements affect the material, encouraging a hands-on approach to learning and discovery.

#### 2.9.3. Focus and Concentration

Crafting and Art activities help children enhance their focus and concentration [29]. The immersive nature of these activities allows them to maintain attention for extended periods, fostering deep engagement and sustained mental effort.

#### 2.9.4. Spatial Thinking

By engaging in three-dimensional modelling, children develop spatial thinking [30]. They start to understand the relationships between different parts of a model and how these parts are arranged in space, which is crucial for grasping complex spatial concepts.

### 2.10. Measuring Results

Measuring the outcomes of a hands-on course for middle-schoolers presents several challenges: mainly, as this study aims to conduct this course without traditional written tests. The lack of formal assessments means we do not anticipate receiving standardized responses from the schoolers. Consequently, we rely heavily on the subjective judgment of the teachers to gauge the schoolers’ reactions and engagement. The teachers record all schooler feedback and quotes during recap sessions following each course.

### 2.11. Limitations

The study faced several limitations, primarily due to resource constraints. With no additional funding available and the regular crafting supplies funded by parents, we could not procure extra materials, which restricted the breadth and depth of the project. Furthermore, modifications to the existing crafting curriculum had to be insignificant, as the educational targets were pre-specified and inflexible. This constraint meant we had to adhere strictly to the established objectives, limiting our ability to innovate or introduce significant changes. Additionally, the absence of written tests, mandated by the curriculum guidelines, precluded us from quantitatively assessing the schoolers’ knowledge acquisition and learning outcomes.

## 3. Results and Discussion

Generally, we were astonished by the schoolers’ lack of knowledge regarding ecology and sustainability, highlighting a significant gap in their understanding of essential environmental concepts such as ecosystems, renewable resources and the impact of human activities on the planet. This gap underscores the critical need for integrating comprehensive ecological education and sustainability principles into the curriculum to foster a deeper awareness and proactive engagement among schoolers in addressing global environmental challenges.

### 3.1. Drawing and First Clay Modelling

During the introductory session, initial scepticism regarding the prospect of working with fungi was noted among the middle-schoolers. The session commenced with a query regarding familiarity with the classic computer game Tetris [31], revealing varying levels of prior exposure. However, the schoolers quickly grasped the concept of constructing Tetris blocks as a limited set of shapes. Each schooler then proceeded to individually draft a scaled 3-dimensional sketch of a selected Tetris block shape on a 1:1 scale with dimensions of 4 cm per edge. This task progressed more swiftly than anticipated due to prior experience with three-dimensional drawings from mathematics classes. Excitement mounted as the schoolers eagerly queued to collect clay to form their positive moulds, yet challenges emerged as they endeavoured to replicate their selected Tetris block shapes precisely. Strategies varied, from assembling individual cubes to moulding complete forms to be gradually reconfigured into the desired Tetris block shape. Guidance was provided to facilitate the clay-forming process and emphasized the need for dimensionally accurate positives that do not interlock with their negatives to ensure ease of separation without damage. Despite considerable progress, achieving precise dimensions (4 cm side lengths) proved elusive within the allocated time, as we expected, necessitating further refinement in subsequent sessions (Figure 1).

Anticipated challenges include maintaining schooler engagement during the meticulous shaping process. Each schooler marked their work for identification before the blocks were collectively gathered, revealing the potential for assembly (as in the Tetris computer game) and the need for further refinement to achieve uniformity. Blocks were prepared for preservation until the next session by sprinkling them with water and wrapping them in cling film. The workshop underscored the diverse skill sets required and the varied approaches schoolers employed to tackle sub-tasks. All schoolers achieved a comparable level of progress by the end of the session. Individual responsibility for workshop cleanup during the concluding minutes was especially stressed.

### 3.2. Finalising the Clay Modelling

The second day of the course, scheduled a week later, posed more significant challenges as the focus shifted towards enhancing the endurance of the schoolers. The objective was to refine the dimensions of the clay Tetris blocks to precisely 4 cm squares. Incorporating three schoolers that had been absent due to illness the previous week added complexity, necessitating their catching-up to align with their peers’ progress. Over the two-hour duration, the schoolers’ motivation gradually declined, varying among individuals. As the schoolers approached the target dimensions, some began to assert that their efforts were satisfactory despite the clay base length exceeding the required measurement by over a centimetre. One schooler, previously recognized for exceptional aptitude in other classes, encountered difficulty achieving precise dimensions, attempting to mask this challenge with a dismissive attitude: “I do not care too much”. Conversely, one of the schoolers who had recovered from illness displayed remarkable engagement, visibly striving to outperform her peers. This heightened engagement could be attributed to her desire to integrate into the leading girl group within the class, having previously experienced discrimination. Half of the schoolers reached a plateau approximately halfway through the session, decreasing further progress. During the final hour, the teachers provided physical assistance to two-thirds of the schoolers to achieve the required precision. A recurring challenge was the thickness of the Tetris blocks, with instances of being either excessively thin or excessively thick. Ultimately, the result was deemed satisfactory, as all Tetris blocks met the predetermined diameter criteria (with the requirement of 0.5 cm left unmentioned). To foster group cohesion, the teachers created two cubes with a base length of 4 cm each, and the schoolers shall integrate these cubes into their project at the course’s conclusion.

The process of collecting and storing the Tetris blocks prompted individuals to spontaneously engage in piecing them together, despite the absence of any requirement to do so (Figure 2). The cleanup process proceeded smoothly, with schoolers demonstrating a willingness to participate despite the need to correct behaviour from two schoolers with a record of being problematic. Notably, these problematic schoolers were among the first to adopt the “good enough“ mentality that was previously observed.

### 3.3. Casting the Negative

The moulding process commenced with constructing a water-tight clay wall around the clay structure, facilitating the pouring of gypsum around the structure. Ensuring a minimum of one centimetre of gypsum above the structure was imperative to construct a closed mould. The procedure was demonstrated using pre-prepared square clay structures by the instructors, with gypsum prepared by thoroughly mixing two parts of water with one part of gypsum. The novelty of gypsum casting elicited excitement among the schoolers, particularly the male cohort, who became eager to progress to the pouring stage. The initiated working speed caused incomplete and leaky clay walls, which led to significant gypsum leakage through holes, adding an element of challenge and excitement as schoolers engaged in identifying and sealing the leaks (Figure 3). The casting and mixing process proceeded sequentially under individual supervision, allowing for the controlled generation of waste. Following casting, the schoolers observed and felt the hardening of the gypsum with keen interest, further heightening their engagement.

The subsequent half-hour (the minimum time needed to clean up the gypsum mess) was dedicated to cleaning procedures for the workspaces and surroundings. The demonstrated gypsum mould was hardened by the session’s end, enabling the extraction of the clay positive and providing a tangible demonstration of the principle of using a positive to create a mould for a positive of another material. This session evoked growing enthusiasm among schoolers for the project, as their understanding deepened.

### 3.4. Cleaning the Negative

Despite the deviation from the original plan due to external circumstances, the extended interval led to the clay hardening more than anticipated. Surprisingly, this resulted in unexpected engagement and excitement among the schoolers, particularly during removal. Using tools like hammers and chisels to remove the hardened clay added an element of fun to the session, which was especially surprising given that the schoolers were from another group (Figure 4). As the schoolers tried removing the inner clay structure without damaging it, their excitement and concentration levels rose, fostering a dynamic and immersive learning environment.

Although two forms suffered minor damage due to having thin walls, this allowed schoolers to learn repair techniques using small quantities of clay. Overall, the unexpected time delay and the resulting challenges added an element of excitement and problem-solving to the session, enriching the learning experience for the schoolers.

### 3.5. Filling the Gypsum Forms with Mycelium

The session commenced with an intensive introduction to establish a clean and disinfected work environment. Lab gloves and face masks were meticulously disinfected using 70% isopropanol and lab tissues, followed by thorough disinfection of the workbenches. The utilisation of masks, serving as an additional precaution against mycelium contamination, sparked reflections on past experiences with COVID-19 pandemic-related protocols, emphasising the importance of these measures in everyday lab environments—the stringent disinfection protocols aimed to safeguard the mycelium from contamination, heightening schooler anticipation. The schoolers (paired into groups) subsequently handled their previously prepared negative forms.

Wrapping disinfected cling film around the forms ensured seamless filling without compromising integrity (Figure 5). As recommended by Grow Bio [4], two layers of cling film were employed to counteract the moisture-reducing effects of the gypsum. This progress point was utilised to synchronise the schoolers’ progress in the session. Following the disinfection of plastic containers, schoolers retrieved blocks of mycelium, which, although they exhibited more significant growth during transport than in our trials, this posed no issue according to Grow Bio. The mycelium was ground manually, with a tablespoon of flour added to provide readily assimilable nutrients during post-grinding stress.

Layer-by-layer filling of the forms ensued, capturing the interest of even the most reserved schoolers. Careful attention was paid to ensure uniform filling, especially along the edges, with periodic reminders emphasising the imperative of maintaining sterility—the final step involved covering the filled forms with perforated cling film to facilitate mycelial respiration.

The palpable excitement and immersion in the lab work were evident, with some schoolers expressing their enthusiasm through personalised name badges. Subsequently, the group naturally divided into two, with one group setting up the mini glasshouse tent for protected mycelial growth while the other engaged in discussions encompassing various scientific disciplines intersecting with the project.

The session concluded with the schoolers cleaning the desks, while floor cleaning was deferred until after their departure to mitigate potential mycelium dispersal. Despite the intense atmosphere, characterised by substantial excitement, the schoolers demonstrated commendable focus on their tasks, making the session manageable for the instructors. Filled forms were stored within view but out of reach in the glasshouse tent, which was augmented with open water containers to enhance humidity levels.

### 3.6. Forming the Communal Mycelium Structure

Cultivation was carried out for six days due to logistical constraints preventing an earlier start. The schoolers wore disinfected laboratory gloves during the procedure. Each schooler received their mycelium block individually. To extract the mycelium from the forms, two-thirds of the forms had to be broken to avoid damaging the mycelium blocks.

As the schoolers handled the mycelium blocks, their reactions varied widely, ranging from disgust to excitement and curiosity. The teachers noted that the mycelium growth was less than expected, potentially due to decreased humidity levels attributed to the gypsum used.

The schoolers sequentially added their mycelium blocks to the communal artistic structure. The individual blocks were pressed together utilizing disinfected toothpicks. The communal structure was returned to the incubation tent for further growth.

The following day, during the scheduled session, we provided a comprehensive introduction to mycelium and its potential applications, emphasizing its role in sustainability. During the presentation, about half of the schoolers showed interest, and some schoolers perceived questions related to biology and physics as unfitting. For the teachers, it was surprising to experience that even basic knowledge of the topic and the broader concept of sustainability was missing.

### 3.7. Educational Results

Understanding positive and negative forms is critical to developing three-dimensional thinking. The children created the clay positive form and cast a negative form (mould) in gypsum; they experienced firsthand how a previously occupied space becomes a cavity. This process enhances their understanding of volume, space and the relationships between dimensions, forming a foundation for creative design and three-dimensional thinking.

The study’s decision to inform the schoolers incrementally about the project’s scope spurred mixed results. A dedicated day would be required to introduce and explain the course adequately. However, this approach does not align with the hands-on nature of the crafting lessons inherent to the course. Additionally, schoolers would not favour a lecture-style format during practical crafting sessions, which was already apparent during the introductory sessions needed during the course. We recommend collaboration with another school course for future courses to transport additional theoretical knowledge.

#### 3.7.1. Skills Developed through Positive and Negative Moulding

Spatial Thinking and Imagination:The schoolers developed or improved their understanding of volume, dimensions and the spatial relationships between objects and shapes.Fine Motor Skills:Working with clay positive forms and gypsum negative forms required precise hand movements, enhancing the fine motor skills necessary for detailed work.Problem-Solving Skills and Logical Thinking:Children learned to identify and overcome challenges in the design process, strengthening their logical thinking and problem-solving abilities.Patience and Perseverance:The modelling and moulding processes required patience and perseverance, teaching the schoolers that creative work demands time and meticulous effort.Self-Confidence and Self-Esteem:Successfully creating works of art and overcoming design challenges boosted the schoolers’ self-confidence and self-esteem.Teamwork and Social Skills:When tasks were given as group work, the schoolers enhanced their ability to collaborate, communicate and share ideas effectively.Cognitive Skills:Understanding and applying positive and negative forms promotes abstract thinking and comprehension of complex relationships.

#### 3.7.2. Holistic Development

The skills acquired through modelling and three-dimensional thinking are crucial for artistic development and contribute significantly to the schoolers’ holistic education and personality development. These activities promote cognitive, social and emotional growth, preparing children for future challenges and opportunities.

### 3.8. Interdisciplinary Results

Interdisciplinary results need to differentiate between the course itself and the development and teaching activities of the teachers. The students faced more difficulties than anticipated in integrating various fields of science: for instance, comprehending the broad and interconnected concepts encompassed by the term “sustainability” proved challenging. This difficulty in transferring knowledge across disciplines highlights a significant gap in current educational practices and underscores the urgent need for more robust interdisciplinary teaching methods in schools.

The interdisciplinary collaboration among the authors (participants) of this study can be evaluated through a more systematic approach. To evaluate participants’ experiences during the preparation and implementation of the course, we designed a questionnaire inspired by a study conducted by the Interdisciplinary and Community Engaged Learning (Educational Development and Training) team at Utrecht University [32]. This study involved interviews with successful interdisciplinary researchers to distil critical factors that contribute to their success [32]. The researchers identified three critical conditions: (1) immersion in the perspectives of others, (2) open-mindedness and modesty and (3) finding common ground.

Participants in our interdisciplinary project were assessed to determine whether they met these criteria. Notably, one participant highlighted that understanding a collaborator’s views in this context involves grasping their content and appreciating the origins of these perspectives. Team dynamics often involved participants with varying levels of expertise, providing opportunities to practice modesty. Despite occasional unmet expectations regarding partners or the project itself, teams effectively managed their tasks and readily found common ground. For another participant, it was important and valuable to occasionally let go of his expectations and accept what comes. Some participants also found that it is important to remain calm and composed when not everything goes according to plan and to trust in your own abilities.

## 4. Conclusions

Integrating mycelium-based engineered living materials (ELMs) into an interdisciplinary middle-school curriculum yields profound insights and educational outcomes. This initiative was spearheaded through a collaborative effort at the University of Applied Arts Vienna and successfully merged art and science, focusing on inspiration and sustainability from Nature. By leveraging the innovation potential of mycelium, schoolers were guided through a transformative journey encompassing design, craftsmanship and scientific inquiry. The workshop facilitated hands-on experiences in creating mycelium-based artwork and instilled a deep appreciation for sustainability practices among participants. Through the meticulous process of designing, modelling, casting and cultivating mycelium, schoolers developed crucial skills in spatial thinking, fine motor skills and problem-solving. These skills are vital for fostering creativity and for preparing schoolers for future challenges in a rapidly evolving world.

The project’s interdisciplinary nature encouraged collaboration and communication across disciplinary boundaries. Schoolers learned to navigate diverse perspectives and embrace the complexities inherent in interdisciplinary teamwork. This holistic approach enriched their educational journey and equipped them with essential skills such as patience, perseverance and self-confidence. Looking ahead, the success of this initiative paves the way for future explorations at the intersection of art, science and sustainability. By continuing to integrate innovative materials and interdisciplinary approaches into curricula, educational institutions can nurture a new generation of creative thinkers and problem-solvers committed to shaping a more sustainable future. Ultimately, this endeavour advances academic scholarship and serves as a testament to the transformative power of interdisciplinary education in addressing global challenges. Through ongoing collaboration and innovation, we can inspire future generations to lead with creativity, empathy and a profound respect for our planet.

## Figures and Tables

**Figure 1 biomimetics-09-00525-f001:**
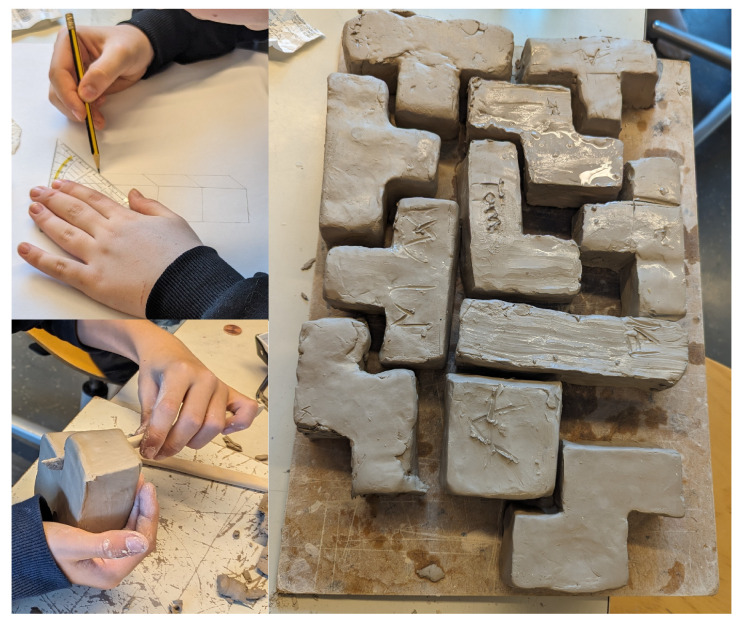
Middle-schoolers drafting and forming Tetris-shaped clay blocks during the introductory session. Challenges with precision and diverse strategies were observed, indicating a blend of creativity and technical skills. Prepared blocks were wrapped for the next session.

**Figure 2 biomimetics-09-00525-f002:**
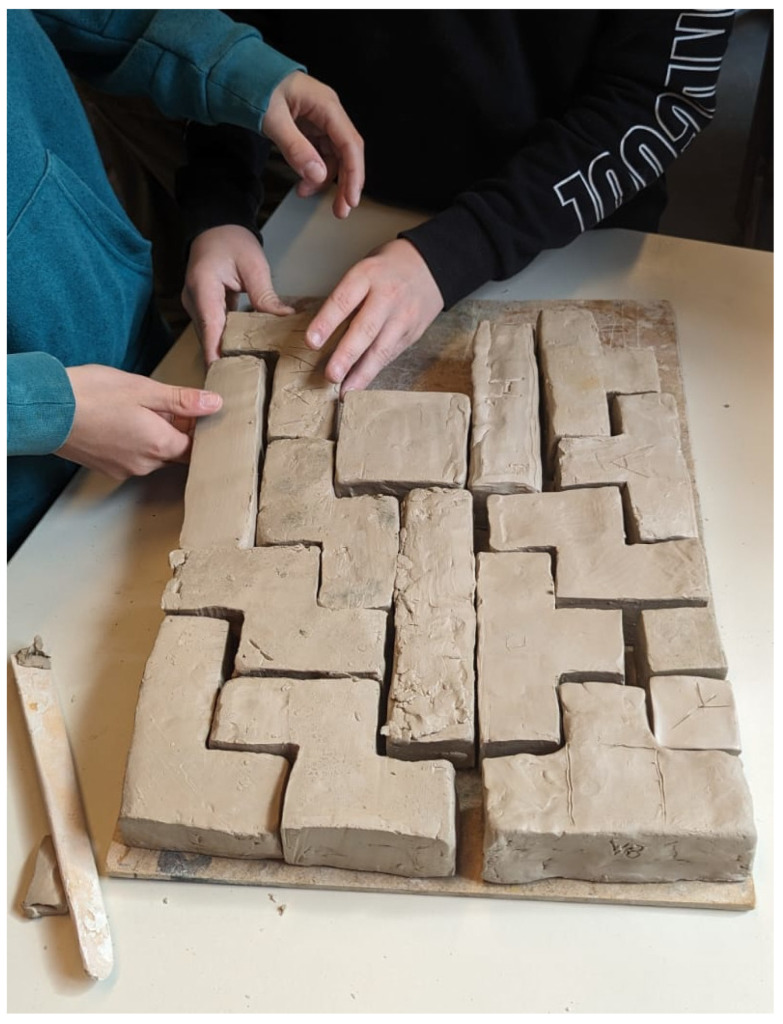
Middle-schoolers refining clay Tetris blocks to precise dimensions with varied levels of motivation and engagement. Teachers provided assistance to ensure all blocks met the criteria, and spontaneous block assembly demonstrated collaboration and group cohesion.

**Figure 3 biomimetics-09-00525-f003:**
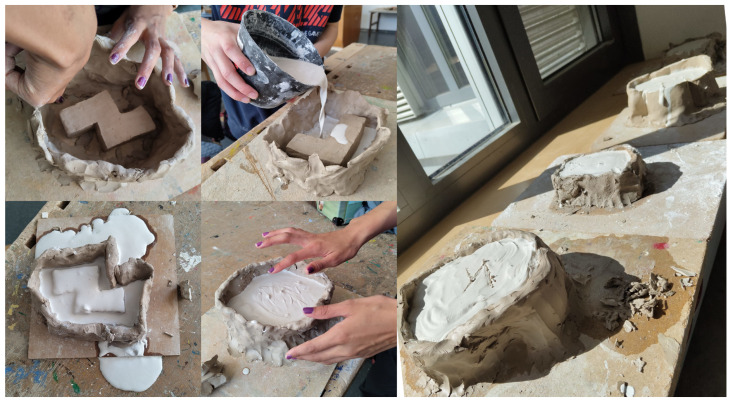
Middle-schoolers constructed clay walls and poured gypsum to create moulds, encountered and solved leakage issues and observed the hardening process, which reinforced key concepts and increased project enthusiasm.

**Figure 4 biomimetics-09-00525-f004:**
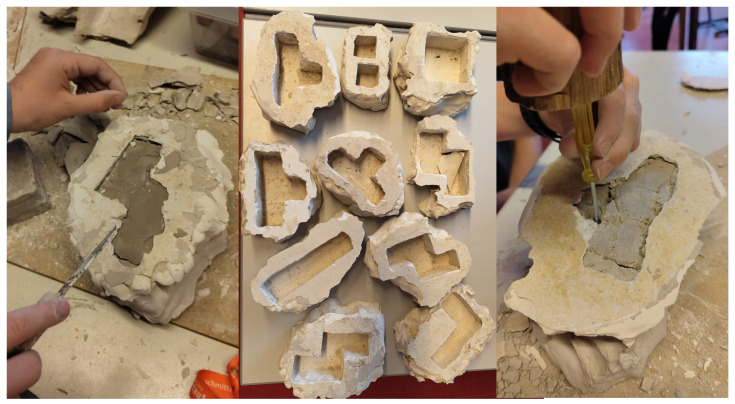
Middle-schoolers remove the hardened clay using tools, demonstrating increased excitement and concentration and learning repair techniques for minor damage.

**Figure 5 biomimetics-09-00525-f005:**
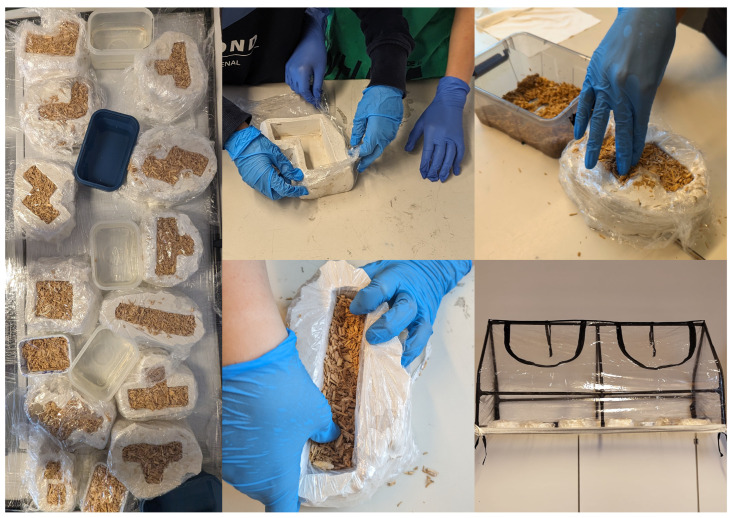
Middle-schoolers prepare and fill forms with mycelium under sterile conditions, integrating practical skills with interdisciplinary scientific discussions.

**Figure 6 biomimetics-09-00525-f006:**
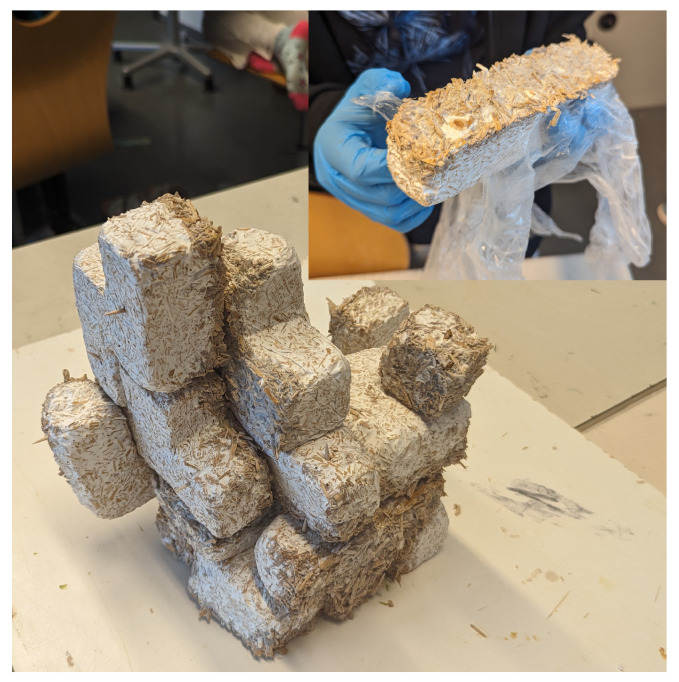
The cultivated mycelium Tetris blocks were integrated into an artistic collective structure and allowed to merge through a renewed growth process.

## Data Availability

All publicly available data is included in the manuscript.

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
