# Peer review of "Harmonizing Nature, Education, Engineering and Creativity: An Interdisciplinary Educational Exploration of Engineered Living Materials, Artistry and Sustainability Using Collaborative Mycelium Brick Construction"

_biomimetics, 2024, doi:10.3390/biomimetics9090525_

Round 1

Reviewer 1 Report

Comments and Suggestions for Authors

The paper describes a compelling exercise that engages students in making molds and growing mycelium.  It excels at describing and illustrating the experience in an accessible way.  It reads as an interesting case study that is a good model for combining craft skills, collaboration and science lessons. 

The paper could be stronger at providing evidence about the educational outcomes of the exercise.  While many areas of learning are mentioned, it was surprising that the paper describes only assessing interdisciplinary collaboration and then does it weakly.  Even though a questionnaire is mentioned, it only includes a few anecdotes about it.  If interdisciplinary collaboration is so crucial, it should be strengthened in the introduction and the paper should include a more thorough analysis of the questionnaire results.

Would there be a way to compare and contrast this project with another more traditional kind of teaching to explain the innovation? If the students were doing another kind of work, would more of them be distracted and less engaged?

In the future, you could give a pre-test and post-test to measure awareness about specific aspects.  A post-project questionnaire could  ask students about their interest and learning.

Making the clay blocks, gypsum molds and growing the mycelium obviously took a lot of effort in terms of materials, tools, facilities and logistics.  To help those interested in replicating this project, could you provide insights into how to streamline these aspects?

The structure of the paper could be improved.

I.                         Introduction

II.                       Mycelium Block Project

III.                    Learning Objectives & Competencies (currently 2.8)

IV.                    Project iteration outcomes (currently 3.1 to 3.6)

V.                      Learning Results

VI.                    Conclusion

Overall, the project looks like a fun and innovative effort.

Comments on the Quality of English Language

Some small language questions:

Lines 26 & 28: children aged 11 and 12 should not be called undergraduates

Line 294: I have no idea what are “personalised breast signs”

Reviewer 2 Report

Comments and Suggestions for Authors

The study aims to contribute to the field of Biomimetic education & sustainability for undergraduate students, through a hands-on artistic session on creating artwork using engineered living materials. The broader goal of this intended scientific publication as stated in the abstract is to ultimately equip educators with a framework/model for interdisciplinary knowledge. Students aged 11-12 engaged in developing communal mycelium brick structures, by first developing flat forms, creating negative forms (moulds) from the given flat forms and finally growing & incubating mycelium brick forms for communal construction from it. An effort is made to elucidate the learning outcomes from different steps of this exercise in the results & discussion section.

Overall, the intent of the study is novel, the work could benefit from clarifying its objectives and providing a clear framework to scale. In the following paragraphs, I have provided specific comments which could help the authors improve the manuscript.

I would like you to consider the following comments.

Introduction:

·       The Title “Harmonizing Nature, Engineering and Creativity: An Interdisciplinary Exploration of Engineered Living Materials, Artistry and Sustainability in Collaborative Mycelium Brick Construction” does not clarify anywhere the manuscript is about teaching approaches for biomimicry/biomimetics through mycelium brick construction. You could consider altering it to reflect an ‘educational approach’ to ‘teach biomimetics through mycelium brick artwork construction’.

·       Line 56- 60: It is a complex sentence, without much clarity.

 Does this promised framework in the manuscript help educators teach students about biomimetics through ELM based artwork or living artefacts? Or is the framework aimed to a be set of steps for mycelium- based artwork construction.

·       Overall, the introduction & abstract raise several questions about the objective of the study.

¨     Is the aim of the said framework to teach about mycelium brick construction, or broadly biomimetics or even more broadly to integrate science & art curricula? The interdisciplinarity is an achievement, but with biomimetics & sustainability being very broad studies in itself, the primary objective & scope of the study can be clarified better?

¨     It seems to the reviewer that the study aims to integrate biomimetics into undergraduate science curriculum by proposing a framework(?) and using the mycelium brick workshop as a case study of the framework(?). Please confirm or deny?

§  Paragraphs on interdisciplinarity & design thinking from Line 61 are not well correlated to the previous paragraphs. Drawing parallels with design thinking as a framework for education is made, but there is no apparent implication that it was used in the study? Can you please justify the parallel drawn in relation to the objective of the study here? Is it a basis to develop a ‘Interdisciplinary Biomimicry teaching approach’?

§  Consider the suitability of the keyword ‘biomimetic’ based on definition and broadening the view to terms such as nature inspired or bioinspired.

§  Please add further literature on

1.     Teaching approaches for biomimicry in school education. E.g - (Coban & CoÅŸtu, 2021).

2.     Similar existing studies done at schools & universities about workshops, coursework, project setups to teach application of biomimicry in an interdisciplinary way.  (Yeter et al., 2023).

Materials & Methods

·       Line 82 – It is rather abrupt, diving into the decisions taken about the choice of workshop building blocks. Consider adding the section ‘Workshop Setup or Structure’ to introduce the title of the workshop, Length and stages of the workshop, the nature of facilitation (by teacher or researcher) and what pre-workshop information was given to the students about the organisms, biomimicry approaches & living artefact design.

·       The section describes the specific workshop steps of growing a mycelium brick but no overarching structured framework like a ‘design thinking framework & stages’ or ‘biomimicry design model’. Absence of a diagrammatic representation of the claimed teaching framework developed is felt. If there is one, please include how it was developed as well.

·        Please include section on participant details and context. Include details on participants’ familiarity with biomimetics/bioinspiration etc., mycelium materials, sustainability etc.

Baseline determination of their knowledge & skills being measured prior to the workshop is important.

·       Consider adding details about ‘expected outcomes (educational & otherwise) for teachers & students’ from the workshop based on the framework.

·       The steps of the workshop are very detailed. Can be reduced to make space for the larger context of teaching biomimicry approaches to follow.

Results & Discussion

·       Sections 3.1 to 3.6 mention several observations about student behaviour but previous sections make no mention of how the data about their levels of engagement with mycelium, the concept of biomimetics and sustainability was measured?

·       What method of qualitative data collection was used here – observation by facilitator, semi structured interviews? surveys?

·       What was the method of qualitative data analysis. There are quotes of the participants in the section. Were the quotes & verbal feedback annotated, coded and a thematic analysis done?

·       It is also unclear what were the parameters defined to judge the efficacy of the workshop in imparting knowledge & skills about biomimetics & sustainability?

 Concluding Remarks

·       The manuscript at various points builds an expectation that it provides a detailed framework for educators to integrated biomimetics & sustainability topics into school curriculum using a workshop on mycelium living artefacts as an example workshop. In the context of teaching, design or business, a framework typically has the following:

o   Goal & scope

o   Principles & values

o   Methods/Processes

o   Roles & responsibilities

o   Tools & Resources

o   Metrics & evaluation

o   Best practices & standards

If there is such a development by the author, providing a diagrammatic overview of the teaching approach in a structured manner could improve the manuscript a lot.

I hope the review could help you improve the manuscript.

Comments on the Quality of English Language

the english is ok

Reviewer 3 Report

Comments and Suggestions for Authors

The concept introduced by the authors is innovative and effective to educate students in multidisciplinary fields. The authors should consider following comments to improve the paper-

1.  Educational results are very qualitative. Quantitative evidence should be provided for various skills/objectives.

2. It is not clear how the authors assess the success of the activities.

3. Limitations of the study should be discussed.

Comments on the Quality of English Language

Need some editing of grammar and spelling mistakes.
